# Virus Infection and Systemic Inflammation: Lessons Learnt from COVID-19 and Beyond

**DOI:** 10.3390/cells11142198

**Published:** 2022-07-14

**Authors:** Aileen Faist, Josua Janowski, Sriram Kumar, Saskia Hinse, Duygu Merve Çalışkan, Julius Lange, Stephan Ludwig, Linda Brunotte

**Affiliations:** 1Institute of Virology, University of Muenster, 48149 Muenster, Germany; a_fais01@uni-muenster.de (A.F.); josua.janowski@ukmuenster.de (J.J.); sriram.kumar@uni-muenster.de (S.K.); s_hins05@uni-muenster.de (S.H.); d.merve.caliskan@gmail.com (D.M.Ç.); j_lang54@uni-muenster.de (J.L.); ludwigs@uni-muenster.de (S.L.); 2CiM-IMPRS, International Max Planck Research School—Molecular Biomedicine, Westfaelische Wilhelms-University Muenster, 48149 Muenster, Germany; 3SP BioSciences Graduate Program, University of Muenster, 48149 Muenster, Germany; 4EvoPAD Research Training Group 2220, University of Muenster, 48149 Muenster, Germany; 5Interdisciplinary Center for Clinical Research, University of Muenster, 48149 Muenster, Germany

**Keywords:** systemic inflammation, viral infection, COVID-19, highly pathogenic avian influenza

## Abstract

Respiratory infections with newly emerging zoonotic viruses such as SARS-CoV-2, the etiological agent of COVID-19, often lead to the perturbation of the human innate and adaptive immune responses causing severe disease with high mortality. The responsible mechanisms are commonly virus-specific and often include either over-activated or delayed local interferon responses, which facilitate efficient viral replication in the primary target organ, systemic viral spread, and rapid onset of organ-specific and harmful inflammatory responses. Despite the distinct replication strategies, human infections with SARS-CoV-2 and highly pathogenic avian influenza viruses demonstrate remarkable similarities and differences regarding the mechanisms of immune induction, disease dynamics, as well as the long-term sequelae, which will be discussed in this review. In addition, we will highlight some important lessons about the effectiveness of antiviral and immunomodulatory therapeutic strategies that this pandemic has taught us.

## 1. Introduction

As zoonotic viruses with diverse reservoirs in their natural animal hosts, both, Influenza A viruses (IAV) and Coronaviruses (CoV) pose a constant and significant pandemic threat to the human population. While endemic strains of IAV and CoV cause recurring seasonal waves of respiratory disease with flu-like symptoms, ongoing intra-host evolution in animal reservoirs, the progressive destruction of natural habitats and climate change along with increased animal trade and consumption are critical factors that increase the chances of human infections with zoonotic viruses, such as highly pathogenic avian influenza A viruses (HPAIV) of the subtypes H5N1, H7N9, or H5N8 as well as the newly emerged pandemic severe acute respiratory syndrome coronavirus Type 2 (SARS-CoV-2), the causative agent of coronavirus disease-19 (COVID-19). Infections can occur by direct contact to infected animals or contaminated environments and can cause severe, often lethal disease.

Confrontation of such non-human-adapted viruses with the highly evolved and multilayered human immune system often leads to an inbalanced activation of the early innate immune response pathways, which facilitates a perturbation in the recruitment of immune cells and their activation [1,2].

While infections with seasonal IAV and CoV are usually restricted to the respiratory tract in otherwise healthy individuals, infections with highly pathogenic viruses are often accompanied by systemic viral replication, secondary bacterial infections leading to sepsis, and detrimental tissue damages in multiple organs [3]. In addition, post-recovery complications such as long COVID can remain for months even in young and immunocompetent individuals. The responsible virus-specific mechanisms of immune activation and evasion are highly diverse and not fully understood. Especially for SARS-CoV-2, these mechanisms along with suitable clinical parameters for the prediction of individual disease trajectories are under intensive investigation by clinics and research groups around the world and have generated remarkable progress during the past two years.

In this review, we focus on the advancements in the understanding of the early immune responses during SARS-CoV-2 infection, identification of prognostic clinical disease markers, and long COVID. We will discuss parallels to infections with HPAIV, summarize the recent developments on the road to disease-tailored treatment strategies that target the virus as well as the host, and finally address the lessons that we have learnt during this pandemic over the past two years.

## 2. Crossing the Species Barrier—Human Infection with Zoonotic Viruses

Zoonotic transmission and establishment of a robust infection by CoV and IAV are spontaneous and sporadic events that are majorly determined by the degree of adaptation to the human receptors, which provides the first species barrier. In addition, the duration and type of contact to the infected animal are decisive. While human–human transmission is still rarely observed in case of HPAIV [4,5,6,7,8] due to a persisting incompatibility of the viral receptor-binding protein Hemagglutinin (HA) to the human type α2-6-linked sialic acid [9,10], the newly emerged zoonosis SARS-CoV-2 already crossed this species barrier by harboring a spike (S) protein that is capable of utilizing the human angiotensin-converting enzyme 2 (ACE2) protein as a receptor [11]. Pre-symptomatic airborne transmission from the upper respiratory tract (URT) as well as the absence of pre-existing immunity within the human population and the unavailability of vaccines or approved antiviral treatments were additional factors that contributed to the rapid virus spread and high mortality in humans in the first months of the pandemic. In order to prevent the collapse of healthcare systems and interfere with viral transmission, rapid enforcement of non-pharmaceutical interventions such as face masks, strict social distancing, contact tracing, and isolation of infected individuals were utilized. Despite these drastic measures, SARS-CoV-2 caused devastatingly high numbers of infections worldwide with severe pneumonia and high mortality in the first months [12]. In an exceptional effort, only one year later, highly efficient mRNA and adenovirus-based vaccines encoding the SARS-CoV-2 S protein were available and together with the nucleoside analog remdesivir, the first antiviral drug with emergency use approval for COVID-19 patients, resulted in a remarkable reduction of fatal cases in countries with access to these measures [13,14]. However, the S protein has demonstrated a high plasticity for continuous evolution which resulted in the emergence of viral variants with improved binding capacities to ACE2, alternative entry mechanisms, and partial evasion from the adaptive immune responses that are evoked by natural infection or vaccination as well as therapeutically used antibodies [15,16,17,18]. These unpredictable developments have resulted in the emergence of numerous subdominant and also several dominant viral variants that displayed individual characteristics of transmissibility and pathogenicity, which still pose a challenge to the clinical management of COVID-19.

## 3. From Local to Systemic—Disease Course and Immune Responses

Disease severity and dynamics of the viral infections are majorly determined by the cell types that express the receptors and thereby dictate the tissue tropism of the virus. CoV and IAV are both airborne viruses that initiate the infection in cells of the human URT and lower respiratory tract (LRT) [19,20]. While seasonal CoV and IAV mainly infect the UTR, SARS-CoV-2 and HPAIV can also infect the LRT, which confers the higher pathogenicity of these viruses. Human lung biopsies of deceased infected individuals and ex vivo infections of human lung tissue have indicated that human lung stem cells in the LTR, known as alveolar type II pneumocytes (AT-II) are a preferred target of SARS-CoV-2 as well as IAV [20,21,22,23,24]. Other cells have also been shown to facilitate viral infection and replication, but infection of LTR cells is associated with severe disease [25,26]. Interestingly, classically activated M1 alveolar macrophages (AM) have been shown to be susceptible to infection by SARS-CoV-2 and contribute to viral spread. Supposedly the lower endosomal pH of activated M1 AMs promotes membrane fusion and virus replication. In contrast, activated AM of the M2 phenotype suppress viral replication by channeling the virus particles to lysosomal degradation [27]. Beyond the cells of the respiratory tract, SARS-CoV-2 replicates systemically and also infects cells in several other organs, including intestinal epithelial cells, endothelial cells, and renal parenchymal cells ([28] and reviewed in [29]), which contributes to the pathogenicity of COVID-19.

Clinical manifestation of COVID-19 is diverse, ranging from asymptomatic patients to severely affected patients and death. Infected individuals can shed virus particles even in a pre-symptomatic period with a peak of infectivity two days before and one day after symptom appearance [30]. Symptoms normally arise after an incubation time of five days and 14 days after exposure in symptomatic individuals [31]. The most common symptoms include fever, cough, fatigue, anosmia, and dyspnea [32,33]. Patients have also reported a sore throat, diarrhea, and nausea [34]. Interestingly, frequently reported symptoms differ between some of the SARS-CoV-2 variant strains. In general, studies suggest a similar symptom range for Alpha-, Beta-, and Delta-infected patients. Additional to the described symptoms for the initial SARS-CoV-2 strain, Alpha patients most commonly suffer from fatigue and headaches [35]. Delta causes a more rapid disease course with higher viral titers on top of specific auditory impairment and gangrene from blood clots [36,37]. In contrast to that, patients that are infected with Omicron suffer more likely from severe fatigue, sore throat, and hoarse voice, but significantly less from loss/altered smell, eye soreness, and sneezing, while hospital admission is also reduced in general [38]. A case study in Europe suggests two disease classifications: on the one hand, individuals with high viral loads in the respiratory tract without severe illness and on the other hand, a two-step progression with decreasing viral loads in the URT but significant worsening of symptoms after around 10 days [39], which develops into severe pneumonia and acute respiratory distress syndrome (ARDS). Severe COVID-19 correlates with systemic viral replication and high blood levels of inflammatory cytokines, which can turn into uncontrolled systemic inflammation that is associated with fever, tachycardia, tachypnea, and hypotension [40,41] as well as extrapulmonary manifestations such as acute kidney injury or thrombosis [42,43,44,45] resembling sepsis characteristics. Consequently, inflammatory manifestations of COVID-19 include cutaneous, hematological, neurological, cardiovascular, renal, pancreatic, endocrine, and ocular involvement additional to the pulmonary damage (reviewed in [46]).

Interestingly, the clinical picture of human infections with HPAIV shares some striking similarities to COVID-19 (summarized in Table 1). Since the first human case in 1997, more than 1700 cases of human infections with HPAIV of the H5N1 subtype with a case fatality rate of approximately 50%, and more than 60 cases of H5N6 infections were reported globally. Only recently, the UK and USA each reported a case of human H5N1 infection in 2021 and 2022, respectively [47]. In addition, more than 1500 infections with low pathogenic avian influenza viruses (LPAIV) of the H7 subtypes (the majority being H7N9) along with H6N1, H9N2 and H10N3, H10N7, and H10N8 were reported until today with symptoms ranging from mild conjunctivitis to severe pneumonia [47]. Although the clinical data on human infections with HPAIV are still very limited, the reported symptoms, clinical manifestations, and risk factors resemble the clinical reports from severe COVID-19. Early symptoms include common flu-like features such as fever, cough, malaise, myalgia, headache, and sore throat, sometimes abdominal pain and diarrhea. Disease progression can be rapid as exemplified by a case report from a H5N6 infection that described how the appearance of fever above 38 °C was followed by hospitalization of the patient and transfer to intensive care unit (ICU) due to dyspnea in a time-frame of two days [48]. Most laboratory-confirmed H5N1 cases are already hospitalized patients with severe complications such as ARDS, pneumonia, and multi-organ failure [49]. In addition, leukopenia and lymphopenia, decreased platelet counts, and internal bleeding due to extensive organ damage, especially in lung tissue, were reported. Other complications included encephalitis and septic shock which eventually lead to death within a median time of nine days post-symptom onset [4,50,51,52,53,54,55,56,57,58]. Similar symptoms were reported for human cases with other avian-derived influenza viruses [59,60]. Similar to COVID-19, several studies report that individuals with severe disease following infections with H5N1 presented elevated concentrations of circulating pro-inflammatory cytokines. These included Interleukin-6 (IL-6), Tumor necrosis factor alpha (TNF-α), Interferon gamma (IFN-γ), Interferon gamma-induced protein 10 (IP-10), and Monocyte chemoattractant protein-1 (MCP-1), which were not only higher compared to individuals that were infected with seasonal influenza viruses but also fatal cases demonstrated higher levels compared to the survivors [52,61,62]. While the number of case reports on human H5N1 infections is rather small compared to COVID-19, the immunopathology of H5N1 infections has been studied in diverse experimental models in vitro, ex vivo, and in vivo. Importantly, the results demonstrate a controversial role of cytokines for viral pathogenesis, which suggests that instead of a general over-activation, the induction of imbalanced inflammatory responses, mediated by virus-specific mechanisms, is one of the leading causes for the pathogenesis of HPAIV in humans [63,64,65,66,67,68]. In line with this, several studies demonstrated that anti-inflammatory therapeutic approaches alone were not sufficient to reduce lethality, suggesting that a combination of antiviral and anti-inflammatory treatments is more suitable [69,70], which has been revealed to be of similar importance for the treatment of COVID-19.

## 4. Biomarkers for the Prediction of Disease Progression in COVID-19 and Infections with HPAIV

Improved understanding of risk factors, prediction of individual disease trajectories, providing information on suitable and disease-targeted therapeutic interventions, as well as the identification of reliable biomarkers and clinical characteristics including immunological and inflammatory proteins, hematological, and organ-specific markers are still the most investigated fields in COVID-19 and IAV research. The description of host genetic and transcriptional markers in experimental infection is relatively common and has revealed promising candidates such as IFI27, which appears to be an early marker also for other respiratory virus diseases to facilitate early infection recognition [71,72]. However, the identification of reliable clinical conditions and biomarkers in patients is by far more complicated. A retrospective study could show an association between initial anemia and increased mortality as well as between a higher ferritin/transferrin ratio and the need for ICU admission with mechanical ventilation [73]. Lymphocytopenia seems to directly correlate with a fatal outcome. Lower lymphocyte counts were found in patients with ARDS, ICU patients, and non-survivors [74,75]. Similar to influenza patients, elevated levels of C-reactive protein (CRP) and also procalcitonin were found to serve as good predictors of severe outcomes [76,77,78]. Elevated levels of the pro-inflammatory cytokine IL-6 could be identified as an important marker for the severity and bilateral lung involvement as well as a predictor of mortality [79,80,81,82]. In addition, high blood levels of IL-1β, IL-2, IL-8, IL-17, G-CSF, GMCSF, IP-10, MCP-1, and TNF-α are indicative for severe COVID-19 [83]. Similarly, levels of D-dimer above 2.0 µg/mL on hospital admission and increased LDH levels were identified to predict mortality [74,84,85]. The results of the MYSTIC study found that the levels of endothelial and glycocalyx markers were indicative for substantial glycocalyx damage that was correlated with a more severe outcome during COVID-19 [86].

Due to the limited number of clinical cases with human HPIAV infections, the identification of solid biomarkers for severe Influenza is less robust. Nevertheless, increased levels of C-reactive protein (CRP) plus the pro-inflammatory cytokines IP-10, CXCL9 (MIG), IL-8, and MCP-1 were higher in patients with H5N1 infections compared to seasonal IAV and associated with severe outcomes [87]. Other manifestations include lymphocyte count, thrombocyte count, and elevated creatinine and aminotransferase levels. A retrospective study of 22 patients from Indonesia also reported elevated D-dimers, CRP, and ferritin levels, revealing impressive parallels to the proposed COVID-19 biomarkers [88,89].

## 5. The Contribution of PRRs to the Innate Immune Responses during COVID-19 and HPAIV Infections

As the first line of defense, the organ-specific innate immune responses are important determinants for the severity of the disease progression in COVID-19 as well as infections with HPAIV [90,91]. Perturbation of these early responses has been demonstrated to significantly contribute to the immunopathology during later disease stages. Increased age and genetic factors that lead to low or abnormal innate immune response are, therefore, considered as risk factors to develop severe disease [29].

The transcriptional programs that are initiated by the host innate immune response in infected cells lead to the expression and activation of cellular proteins that limit virus replication, induce cell death, and warn neighboring cells to curtail viral spread. Intracellular pattern recognition receptors (PRRs) in epithelial and immune cells identify pathogen-associated molecular patterns (PAMPs) at different steps during the virus life cycle and initiate signaling cascades that lead to the induction of interferons (IFNs) together with diverse mediators of inflammation [92]. The different families of PRRs, including RIG-I-like receptors (RLRs), nucleotide-binding oligomerization domain (NOD)-like receptors (NLRs), and Toll-like receptors (TLRs) are expressed on the cell surfaces as well as intracellular compartments and display individual binding preferences for pathogen-derived nucleic acids or proteins. Downstream signaling involves the two major adaptor proteins, myeloid differentiation primary response 88 (MyD88) and TIR-domain-containing adapter-inducing interferon-β (TRIF) that further activate different transcription factors such as NFκB and interferon regulating factors (IRFs) that regulate the expression of pro-inflammatory cytokines such as IL-6, IL-1β, and IL-8, in addition to type I, II, and III IFNs and TNFs (Figure 1) [93]. The nature of the induced antiviral responses in an infected cell is largely determined by the cellular equipment with PRRs, the type of PAMP, the affinity of their receptors, and the virus-specific mechanisms to counteract recognition and signaling [94]. Emerging data also suggest an important role of AMs in the pathogenesis of COVID-19. However, reports provide a mixed picture of the permissiveness and functional contribution of AM to COVID-19 and studies are difficult to compare due to the application of different experimental models or readouts. Nevertheless, there seem to be unique features of macrophages of the M0, M1, and M2 phenotypes during COVID-19. Using a model of human pluripotent stem cell -derived macrophages Lian et al. demonstrated that these cells are not permissive to SARS-CoV-2 but are activated in response to antibody-mediated uptake of infected cells. The differences in the activation were suggested to derive from the expression of unique sets of PRRs that react to the recognition of virus-associated PAMPs [95]. Some studies argue that M1 macrophages upregulate inflammatory factors upon infection [95], while other studies showed that challenged macrophages from bronchoalveolar lavage fluid with SARS-CoV-2 are incapable of producing IFNs suggesting that viral RNA is not sensed [96]. During influenza infections, AMs are susceptible to abortive infection and efficiently sense the release and replicated cytoplasmic viral RNA and the M2 protein. Consequently, the production of Type I IFNs, CXCL5, CXCL9, CXCL10, CXCL11, TNF-α, and members of the IL-1 family is upregulated [97].

Recent studies provided evidence for the contribution of RLRs and some TLRs as cytoplasmic sensors of viral PAMPs for the development of severe COVID-19 and infections with HPAIV [98,99,100,101,102]. While RIG-I was identified as the major sensor for IAV RNAs [103], SARS-CoV-2 infection is preferentially sensed via the recognition of viral intermediates by melanoma differentiation-associated protein 5 (MDA5)and Laboratory of genetics and physiology 2 (LGP2) in epithelial cells [104]. These differences are determined by the individual viral replication strategies within different cellular compartments and the different nature of the viral genomes. The interaction of activated RIG-I with the protein mitochondrial antiviral signaling (MAVS) induces TNF receptor-associated factor (TRAF), IκB kinase (IKK), TANK-binding kinase 1 (TBK1), and finally the transcription factors IRF3, IRF7, and NFκB for the induction of type I and type III IFNs (reviewed in [105]). By autocrine and paracrine binding to their respective receptors, IFNs activate the Janus kinase (JAK)/signal transducer and activator of transcription protein (STAT) pathway, which leads to increased levels of PRRs and the expression of interferon-induced genes (ISGs), such as the viral restriction factors Myxovirus resistance protein 1 (MxA), 2′-5′-oligoadenylate synthetase 1 (OAS1), and ISG15 that inhibit virus replication [94].

The contribution of these mechanisms to the excessive induction of pro-inflammatory cytokines during HPAIV and SARS-CoV-2 infections is widely discussed. A recent report suggested that especially aberrant viral genomes, consisting of the 3′- and 5′-promoters but harboring large internal deletions, which are produced by the non-adapted viral polymerase of avian-derived IAV viruses in human cells act as potent immune stimulators that contribute to cytokine overexpression [106]. In a pre-published report, a similar mechanism was proposed to occur upon SARS-CoV-2 infection. Here, an incompatibility of the viral polymerase with the host transcriptional machinery was proposed to result in the production of higher levels of partially double-stranded small viral RNAs (svRNAs) encoding the 5′ ends of positive-sense genes compared to the human-adapted CoV OC43 and 229E. svRNAs were a potent ligand for RIG-I and resulted in the expression of IFN-β [107].

Interestingly, higher expression levels of TLR1, TLR4, TLR5, TLR8, and TLR9 and the TLR-adaptor protein MyD88 were also positively correlated with the severity of COVID-19 [108]. In addition, in vivo experiments in TLR2-deficient mice revealed that TLR2 activation occurred by the SARS-CoV-2 envelope (E) protein and contributed to IL-6 induction [108]. Accordingly, another study supported the importance of the TLR2 and NFκB axis during SARS-CoV-2 infection [109]. Thus far, the role of other TLRs in COVID-19 is still under investigation. TLRs, majorly TLR3 and TLR7, are also essential for the induction of the innate immune response against influenza viruses especially in immune cells, however, a distinct role in the sensing of HPAIV infections is not well established (reviewed in [110]).

In addition to RLRs and TLRs, the RNA-dependent protein kinase R (PKR) plays an important role for the innate immune response during IAV infections. It is reported that PKR is activated during viral infection by sensing short stretches of double-stranded viral RNAs, which leads to the inhibition of the cellular translation initiation factor eIF2-α and facilitates a host translational shut-off to restrict viral protein synthesis [111]. In contrast to this mechanism, other studies suggest a role of PKR in the induction of overshooting IFN and cytokine levels by HPAIV. Krischuns et al. demonstrated that the replication of HPAIV activates the signaling cascade PKR/p38/MSK1, which facilitates phosphorylation of the host transcriptional co-repressor TRIM28 at serine 473. Intriguingly, this modification alleviates the co-repressor function of TRIM28 and leads to increased levels of IFN-β, IL-6, and IL-8 in lung epithelial cells [64]. Whether PKR activation is facilitated by viral RNAs, possibly the discussed aberrant vRNAs of HPAIV, remains to be shown. While it was suggested that PKR activation during SARS-CoV-2 infection contributes to the delayed onset of the IFN response, it is unknown whether there are additional mechanisms of PKR that contribute to the high induction of pro-inflammatory cytokines [112].

## 6. Adaptive Immune Response against SARS-CoV-2

While T-cell responses are generally developed early within 6-10 days after exposure to primary SARS-CoV-2 infection and protect against severe disease, delayed onset of T-cell responses due to imbalanced innate immune reactions are associated with high levels of IFNs and other cytokines and are correlated with severe clinical outcomes and death [113,114]. Nevertheless, critically ill patients do not necessarily display a stronger T-cell response compared to patients with mild disease [115]. A number of studies report a dysregulation of B- and T-cells together with a strong T-cell lymphopenia in more severe cases of COVID-19 compared to moderate cases [114,116,117]. Although an increasing number of studies provide insights into adaptive immune responses during COVID-19 (reviewed in [118]), there is still a need for a better understanding of the mechanisms underlying B- and T-cell activation. Although the development of new variants reduced the vaccine efficacy which facilitates increased risk of reinfections, the vaccine- and infection-induced T-cell responses seem to be widely retained over a long period of time and provide a protection against severe disease and death. Additionally, previous infection with SARS-CoV-2 provides 84% lower risk of reinfection for an average of seven months [119], enforcing the important role that the adaptive T-cell response plays for the protection against COVID-19. 

## 7. Importance of Interferons for COVID-19

IFNs serve as the primary responders against virus infections. However, beyond the coordination of antiviral actions, induction and signaling of type I IFNs were also shown to play a crucial role for inflammation and pathology in COVID-19. Upon binding to their receptors (IFNAR1/R2) on the cell surface, they activate the JAK-STAT signaling and induce expression of ISGs, which orchestrate the antiviral innate-immune responses. Although during viral infections IFNs primarily exhibit an antiviral function, they also shape inflammatory responses by their immunomodulatory effects on the activation status of immune cells. Analysis of the systemic IFN signatures of hospitalized COVID-19 patients revealed that the cell-specific responses were associated with distinct IFNs [120]. While the transcript signatures in circulating immune cells reflected antiviral responses that were associated with IFN-α2 and IFN-γ signaling, the proteome signatures revealed patterns of platelet activation and endothelial damage that were closely correlated with responses that were induced by IFN-α6 and IFN-β. In addition, IFN-γ and IFN-β levels were associated with high CRP levels as a prognostic marker for poor outcome as well as an increased ratio of neutrophils to lymphocytes as a marker of late severe disease, respectively [120]. The observed differences in the IFN landscape were linked to clinical implications such as seroconversion and hospitalization time, corroborating the importance of an intact IFN response to prevent the development of severe COVID-19 and death. Also, during infections with IAV an intact IFN response is crucial [51,90].

Individuals with inborn errors of type I IFN innate-immunity impacting IFN secretion or IFN response were reported to suffer from severe viral diseases in either childhood or early adulthood [121]. Three inborn errors of immunity: the functional deficiencies of transcription factors IRF7, IRF9, as well as the RNA receptor TLR3, were shown to promote influenza-associated pneumonia [122,123,124,125]. New insights were provided by the COVID Human Genetic Effort (COVID-HGE), which revealed that more than 3.5% of patients with severe COVID-19-associated pneumonia carried previously described deficiencies in *IRF7* and *IFNAR1*, or *TLR3*, *TICAM1*, *TBK1*, and *IRF3* [126] (Figure 1). In addition, this study identified several novel genetic mutations in *UNC93B1*, *IRF7*, *IFNAR1*, and *IFNAR2* leading to life-threatening deficiencies in the IFN response [126]. A chromosome-wide genetic approach uncovered X-linked recessive TLR7 deficiency as a risk-factor for life-threatening severe COVID-19 in <60 years old men [127]. While endosomal TLR7 is long known to sense single-stranded RNA (ssRNA) [128], and its coding sequence has been under strong negative selection [129], its exact roles in human innate immunity remained enigmatic for years [125]. These recent discoveries highlight the essential roles of double-stranded RNA sensor TLR3, ssRNA sensor TLR7, and type I IFN innate immunity in restricting SARS-CoV-2 infection. Interestingly, none of the patients with specific deficiencies of the type III IFN cascade (IL10RB deficiency) suffered from life-threatening viral infections, including COVID-19 pneumonia [126].

### 7.1. Type I IFN Autoantibodies in COVID-19

Autoimmune B-cell phenotypes in humans exhibiting inborn errors of cytokine immunity can produce neutralizing autoantibodies (Auto-Abs) against IFNs such as IFN-α,β,ω (favoring viral diseases), IFN-γ (favoring mycobacterial diseases), or against cytokines such as IL-6 (favoring staphylococcal diseases) and IL-17A, IL-17F (favoring mucocutaneous candidiasis), that resemble the clinical phenotypes of mutations encoding these defective cytokines and/or their receptor subunits (reviewed by [130]). Type I IFN Auto-Abs were initially reported in patients that were diagnosed with systemic lupus erythematosus, myasthenia gravis, thymic abnormalities, as well as in IFN recipients (reviewed in [108]). While these type I IFN Auto-Abs received little attention due to the absence of negative clinical reports [125,130] the COVID-19 pandemic has put a new spotlight on the immunopathological implications of type I IFN Auto-Abs for human susceptibility to viral infections and disease progression. More than 10% of patients with severe COVID-19-associated pneumonia tested positive for neutralizing Auto-Abs against IFN-α2 and/or IFN-ω [131]. Around 94% of infected carriers were men, and almost half of them aged 65+ years, thereby establishing first-line evidence that the higher prevalence of type I IFN Auto-Abs in males and older individuals explain their high risk to severe COVID-19. These observations were successively replicated via autonomous cohorts by other studies [132,133,134,135,136,137,138,139,140,141,142,143,144,145,146]. Further immunogenetic analysis revealed that all carriers of IFN-α2 Auto-Abs also had Auto-Abs against the other IFN-α subtypes (IFN-α1, 2, 4, 5, 6, 7, 8, 10, 13, 14, 16, 17, 21), while only two of these carriers had Auto-Abs against IFN-β, one against IFN-κ, and two against IFN-ε [131]. The carriers of neutralizing Auto-Abs against the IFN-α subtypes had low or undetectable plasma-levels of the 13 IFN-α subtypes during the disease course [131,147,148]. Interestingly, type III IFN Auto-Abs are only rarely detected in the severe COVID-19 cohort [131,149]. Not surprisingly, SARS-CoV-2-infected individuals with known preexisting Autoimmune Polyglandular Syndrome type I (APS-1) and type I IFN Auto-Abs had a high risk of progressing to severe COVID-19 [131,150,151]. In a clinical study of 22 APS-1 patients, 19 patients progressed to severe COVID-19, of which four died, while other patients presented asymptomatic or paucisymptomatic infections, likely due to earlier medical intervention [152]. Interestingly, another clinical report described four young APS-1 patients with type I IFN Auto-Abs that only presented mild-to-moderate COVID-19 [153]. However, a large cohort with 34,000 individuals aged between 20 and 100 years revealed a striking prevalence of neutralizing Auto-Abs against IFN-α and/or IFN-ω with increasing age also in the normal population [131,146]. Type I IFN Auto-Abs were shown to not only diminish the circulating levels of type I IFNs but also to reduce their early local expression in the nasal epithelium leading to compromised antiviral barrier in the URT [154] (Figure 1). It is highly likely that type I IFN Auto-Abs also affect other viral infections, such as HPAIV infections. A recent report suggested that type I IFN Auto-Abs contributed to the development of adverse reactions following immunization with live attenuated yellow fever virus vaccine [155], thereby qualifying their presence/absence as a decisive factor for the safety evaluation of prophylactics and therapeutics.

Taken together, these new insights have important real-life implications and recommend close monitoring and early vaccination of individuals with known defects in the IFN responses to reduce the risk of severe COVID-19 and other similar virus diseases. In addition, such patients should be restrained from donating convalescent sera for clinical studies and therapeutic applications as the transfer of type I IFN Auto-Abs could have severe consequences [131,156]. These findings further help in fine-tuning prophylactic and therapeutic strategies, including plasmapheresis, plasmablast-depleting monoclonal Abs, and targeted inhibition of type I IFN-responsive B-cells [131,157]. Furthermore, as early treatment with IFN-α2 is likely not beneficial to this patient cohort, the therapeutic potential of nebulized IFN-β could be evaluated, as anti-IFN-β Auto-Abs are rarely reported in individuals with type I IFN Auto-Abs.

**Figure 1 cells-11-02198-f001:**
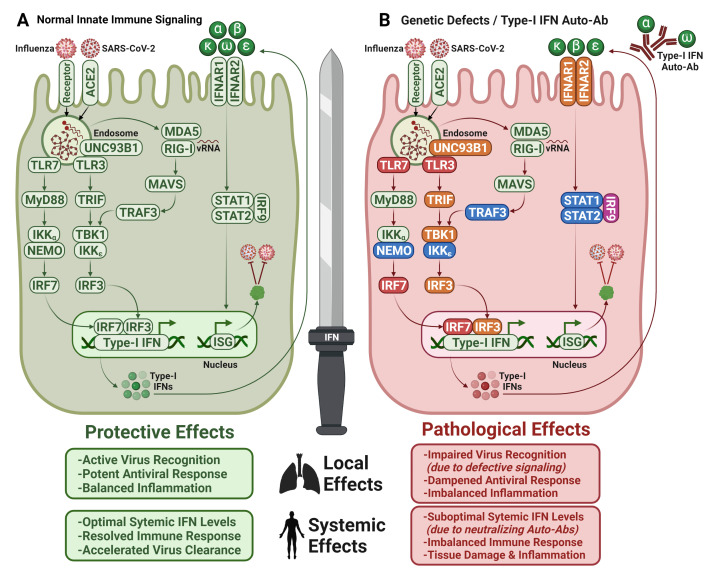
**Type I IFNs—a double-edged sword.** (**A**) Host proteins and signaling pathways that are involved in virus recognition and antiviral response in healthy individuals. (**B**) Defective IFN-dependent antiviral response in individuals with inborn genetic errors resulting in impaired virus recognition or suboptimal circulating IFN levels due to neutralization by type I IFN Auto-Abs. Non-affected proteins (green), defective proteins that are associated with COVID-19 (red) or influenza pneumonia (orange), SARS-CoV-2 and IAV infections (pink), other viral infections (blue); Normal signaling (green arrows), dampened signaling (red arrows). ACE2: Angiotensin-Converting Enzyme 2, TLR: Toll-Like Receptors, MyD88: Myeloid Differentiation Primary Response 88, IKK: Inhibitory Kappa B Kinase, NEMO: NFKB Essential Modulator, IRF: Interferon Regulatory Factor, TRIF: TIR-domain-containing Adapter-inducing Interferon-β, TBK1: TANK-Binding Kinase 1, UNC93B1: Unc-93 homolog B1, MDA5: Melanoma Differentiation-Associated Protein 5, RIG-I: Retinoic acid Inducible Gene I, MAVS: Mitochondrial Antiviral-Signaling Protein, TRAF: TNF Receptor–Associated Factor, IFNAR: Interferon Alpha Receptor, STAT: Signal Transducer and Activator of Transcription, ISG: Interferon Stimulated Genes. Figure adapted from [125,126,131,158] and created with BioRender.com.

### 7.2. Therapeutic Application of IFNs for COVID-19 and HPAIV Infections

Type I and type III IFNs exert strong antiviral activities by inducing ISG-mediated antiviral effectors at the cellular level. In addition, they enhance the functions of monocytes and macrophages, promote CD4^+^ and CD8^+^ T-cell and B-cell responses, and enhance the actions of dendritic cells and natural killer cells. Recombinant IFN-α2 is the only clinically-approved subtype for antiviral treatment against chronic hepatitis B and hepatitis C virus infections, while IFN-β is licensed for treatment of Multiple Sclerosis [159,160]. On the other hand, type I IFNs can exert strong immunopathological activities, by either inducing immunosuppressive effects that interfere with antiviral responses, or by promoting histopathological inflammation that aggravates disease [161]. At the systemic level, type I IFN treatment induces common side-effects such as chills, fever, myalgia, and headache, which are rarely dose-limiting unlike the other uncommon adverse side-effects such as hematotoxicity and neurotoxicity [162]. Despite these risks, the reported virus-induced delay of the IFN response and the high sensitivity of SARS-CoV-2 to exogenously applied type I IFNs encouraged several clinical trials to be conducted during the early pandemic [163,164]. However, the outcome of systemically applied type I IFNs in severely ill COVID-19 patients was rather disappointing and suffered from the lack of a clear benefit as well as the application of additional co-treatments which hampered the analysis of IFN-only benefits. In contrast, the data for the therapeutic use of IFN-λ are more promising. As IFN-λ receptors are localized only along the gastrointestinal and respiratory epithelium, IFN-λ subtypes exhibit lesser inflammatory effects but stronger antiviral activities than the other IFNs, which may provide an explanation for their greater therapeutic benefit [165]. The antiviral potential of IFN-λ subtypes against SARS-CoV-2 were shown in vitro, and were evident from the ILIAD trial (NCT04354259) in ambulatory uncomplicated COVID-19 patients [166,167,168,169]. As part of the TOGETHER trial, recent results using a single subcutaneous shot of pegylated IFN-λ (type III IFN) in vaccinated, non-hospitalized patients within the first seven days after symptoms onset demonstrated up to 50% protection against hospitalization and 60% against death [170]. However, publication of the results is still awaited.

Our current knowledge on the exogenous administration of recombinant IFNs for prophylaxis/therapy against HPAIV infections is only limited. A single-time low-dose IFN-α2 pretreatment significantly reduced the pulmonary viral titers in H5N1-infected mice, while the antiviral effect was improved with multiple pre-treatments, indicating that even a low-dose IFN-α treatment induces a potent antiviral program that reduces virus titers in lungs [171]. To date, only one double-blind, placebo-controlled Phase-II clinical trial (NCT00895947) investigated the prophylactic effects of oral, low-dose IFN-α administration against respiratory diseases, including influenza, in healthy adults, and reported that the treatment alleviated disease symptoms, especially in vaccinated individuals, though it was ineffective in preventing virus-infections [172]. These studies strongly support the high potential of IFN-α for prophylaxis or therapeutic treatments against IAV infection. However, solid and comprehensive research to determine the antiviral properties of human IFN-α subtypes against seasonal and highly pathogenic influenza viruses in a human study model is not available and needs to be performed to enable clinical applications. The suboptimal therapeutic effects of IFN-α2 and IFN-β encouraged researchers to explore the functional diversity of IFN-α subtypes for antiviral treatment against SARS-CoV-2 and influenza viruses [173,174,175,176]. These reports demonstrate the induction of distinct and subtype-specific transcriptomic landscapes which translate into virus-specific antiviral properties in different tissues, further suggesting their individual therapeutic potential [177].

## 8. Antiviral and Immunomodulatory Treatments for COVID-19

The therapeutic approaches against COVID-19, to a great extent, resemble the strategies that are employed against infections with seasonal IAV and HPAIV and include the inhibition of viral entry, blocking of the viral enzymes, and targeting of virus supportive host factors to restrict viral replication. In addition, immunomodulatory strategies to reduce or rebalance the exaggerated and uncontrolled immune responses were widely investigated. As for IAV, a high risk of resistance development is associated with use of direct-acting antivirals during COVID-19, which needs to be tightly controlled. Due to the unavailability of approved antivirals, the use of neutralizing antibodies from convalescent sera was one of the first approaches to be investigated for antiviral therapy of COVID-19 [178]. This technique provides immediate short-term immunization against infectious agents by transferring virus-specific neutralizing antibodies and has been successfully applied for other highly infectious viral diseases such as the Spanish flu, SARS in 2003, pandemic influenza A (H1N1) in 2009, HPAIV, and Ebola by limiting virus replication in the acute phase of infection and enabling rapid recovery [179,180]. While the Food and Drug Administration (FDA) granted emergency use for convalescent plasma therapy for COVID-19 in August 2020, this decision was revised on 4 February 2021, only recommending plasma with high neutralizing antibody titers for use in hospitalized patients in early disease phases or in patients with humoral disorders [181]. In the past, convalescent plasma therapy was mostly only used in epidemics and pandemics. Its effectiveness can be correlated to the pathogen as well as the timing and dosage of the treatment [182]. More controlled studies are required to fully evaluate the clinical effectiveness of convalescent plasma including its use in immunocompetent patients with severe disease. In addition to convalescent therapy, neutralizing monoclonal antibodies (mAbs) targeting the receptor-binding site of the SARS-CoV-2 S protein were rapidly identified [183] and tested in clinical studies (RECOVERY, REGEN-COV), either alone or in combination [184,185]. Similar to convalescent sera, the use of neutralizing mAbs is preferably useful in immunocompromised hospitalized COVID-19 patients but not recommended for broad applications due to the high risk of resistance developments. In addition, the effectiveness of mAbs suffers from rapid evolution of SARS-CoV-2 variants, which present diverse degrees of immune escape from infection and vaccine-elicited antibodies along with reduced sensitivity towards the available therapeutic mAbs. While the mAB sotrovimab demonstrates stable neutralizing activity against Omicron BA.1 and BA.1.1, it’s in vitro activity against the Omicron BA.2 subvariant is already significantly reduced. Currently, bebtelovimab retains high in vitro activity against circulating Omicron subvariants. Treatment guidelines recommend mAb therapy only if paxlovid or remdesivir is unavailable [186].

### 8.1. Direct-Acting Antivirals for Treatment of COVID-19

Remdesivir is a nucleoside analog prodrug, which is metabolized intracellularly to an adenosine triphosphate (ATP) analog that inhibits the activity of the viral RNA polymerase (sold under the brand name Veklury, Gilead) [187]. Its function as a broad-spectrum viral RNA-dependent RNA polymerase (RdRp) inhibitor is well studied and it is effective against several positive and negative-sense RNA viruses including Ebola, hepatitis C virus, respiratory syncytial virus, as well as SARS-CoV-2 in vitro and in vivo [188,189]. In contrast, it only shows low antiviral activity against the segmented negative-sense RNA viruses IAV, Lassa Virus, and Crimean-Congo hemorrhagic fever virus, which is likely facilitated by differences in the structural architecture of the polymerase active sites which disfavors remdesivir as a substrate for RNA synthesis [190,191,192]. So far, remdesivir is effective against all SARS-CoV-2 variants and the only antiviral drug that is approved for COVID-19 treatment by the FDA for hospitalized patients with a high risk to develop severe COVID-19 [186]. Only recently, the FDA has extended the approval of remdesivir also for the treatment of non-hospitalized adults and pediatric patients with mild to moderate disease that are at high risk to develop COVID-19 based on a randomized, placebo-controlled clinical trial (NCT04501952) [193]. Alternatively, the orally available nucleoside analog molnupiravir (sold as Lagevrio, Merck) has received emergency use approval by the FDA in December 2021 for non-hospitalized adults with mild to moderate COVID-19 at high risk to develop severe disease within five days of symptom onset in the absence of access to different antivirals (EUA 108 Merck Molnupiravir FS for HCPs FS Lagevrio 03232022 (fda.gov)). In contrast to remdesivir, molnupiravir is an orally available drug that reduces viral replication by a mutagenesis mechanism, which fosters the general concern of enhanced resistance development and putative integration into the human DNA [194]. Therefore, relevant guidelines recommend that molnupiravir is only used in the absence of access to alternative antivirals such as remdesivir and should be restricted to five consecutive days [186]. Despite the substantial differences in the virus biology, the effectiveness of other nucleoside analogs to inhibit the activity of viral polymerases and restrict viral replication was also shown against several other viruses, including the 2009 H1N1 pandemic influenza virus and subtypes H5N1, and H7N9 [195,196]. In vivo studies show that the drug favipiravir improved the survival of HPAIV-infected mice [197] and was also considered as an antiviral against COVID-19 with ongoing clinical trials. Unfortunately, no significant improvement of the clinical parameters or recovery rate could be observed yet, neither in mild nor in moderate cases of COVID-19 [198,199]. Nevertheless, treated patients showed a tendency to resolve earlier from fever and cough [198]. In light of additional emerging zoonoses with high pandemic potential, the virus and disease-specific applications of nucleoside analogs should be intensively studied. 

Main protease (Mpro or 3CLpro) and papain-like protease (PLpro) are viral proteases that are required for the replication of SARS-CoV-2. However, targeting these crucial steps by lopinavir/ritonavir and darunavir/cobicistat did not provide convincing clinical benefit in patients with COVID-19 [200]. In contrast, nirmatrelvir, an orally-administered C3-like protease inhibitor that is sold in combination with ritonavir under the brand name Paxlovid, is recommended for use in high-risk, non-hospitalized COVID-19 patients from the age of 12 within five days after symptoms onset. However, due to extensive drug interactions, the application of this drug is restricted [186,201]. Unfortunately, studies using recombinant Mpro have shown that certain amino acid substitutions are associated with reduced activity (G15S, H164N, H172Y, and Q189K; 4- to 233-fold reduction) [202]. P132H mutation in nsp5 (Mpro) has been reported in the novel Omicron variant, harboring the risk of resistance [203].

### 8.2. Immunomodulatory Strategies to Alleviate Immunopathology in COVID-19

About 10-20% of COVID-19 patients develop severe symptoms with systemic inflammation as the second stage of disease, severe lung infection, multi-organ failure, and diffuse/disseminated intravascular coagulation following severe pneumonia [204,205]. Previous experiences from influenza pandemics suggested that therapeutic approaches for COVID-19 require antiviral as well as anti-inflammatory strategies. Although the peripheral blood cytokine profiles are not identical, the key mediators of immunopathology are common to both lethal H5N1 influenza infection and COVID-19. Especially the pro-inflammatory cytokines IL-6, IL-1β, TNF-α, IL-10, and IP-10 have been shown to correlate with disease severity in both infections, thus indicating that similar host signaling processes are involved in the disease development. Hence, the immunomodulatory strategies for both viral infections share high similarity. Beyond broad immunosuppressive strategies using corticosteroids that have demonstrated contradictory benefits during progressed and severe IAV infections and COVID-19, more targeted strategies to block or redirect cytokine-specific immune response pathways are intensively investigated and demonstrated promising effects. Especially strategies to reduce the concentration and signaling of harmful cytokines such as IL-6, IL-1, and TNF-α have been intensively investigated in clinical studies for COVID-19.

The IL-6 receptor-directed mAb tocilizumab is specifically directed to the membrane-bound IL-6 receptor (mIL6R) and soluble IL-6 receptor (sIL6R). Clinical studies have shown conflicting results regarding the use of tocilizumab in the treatment of severe COVID-19 [206,207]. However, it has been reported that tocilizumab reduces mortality and the need for mechanical ventilation in severe COVID-19 patients [208,209]. IL-6 is involved in a number of essential anti-viral defenses, including CD8^+^ T cell function and differentiation, T-cell responses, macrophage activation, and migration. Therefore, it should be noted that the use of these drugs targeting IL-6 in the early stages of COVID-19 may result in inhibition of the following antiviral defense steps [210]. Although the risk of bacterial infection due to this immunosuppression is considered, WHO and NIH guidelines recommend the use of IL-6 receptor blockers for the treatment of COVID-19. IL-1α is quickly expressed upon lung cell necrosis and triggers the synthesis of IL-6, TNF-α, Granulocyte-macrophage colony-stimulating factor (GM-CSF), and IL-17, thus the IL-1 blocker anakinra was suggested as an alternative treatment for patients in which corticosteroid treatment, also combined with tocilizumab, was not beneficial. Mortality rates and hospitalization time could indeed be reduced [211,212].

TNF-α triggers the cytokine release syndrome and facilitates the interaction of ACE2 with SARS-CoV-2. The studies on the use of TNF-α inhibitors for the treatment of severe influenza have been a guide for the treatment of COVID-19. High TNF-α serum concentration is associated with severe COVID-19 disease, and it has been speculated that anti-TNF therapy can be used in high-risk elderly patients with COVID-19 [114,213]. Interestingly, COVID-19 patients who regularly use anti-TNF agents due to inflammatory bowel diseases (Crohn’s disease or ulcerative colitis) generally had a mild disease. This suggests that anti-TNF therapy indirectly reduces the overshooting response of the immune system in patients with COVID-19 [214,215,216]. However, the time of administration, dose, and stage of the disease are critical for TNF-α blockers, and early use may accelerate viral replication and worsen the clinical course of the disease, as in other immunomodulatory treatments [217,218]. Anti-TNF strategies have not been useful in the treatment of inflammatory conditions, such as sepsis, and clinical trials were not conducted in humans neither for HPAIV infections nor for COVID-19 [217].

JAK inhibitors such as baricitinib or tofacitinib are recommended as a treatment for COVID-19 as they can prevent the phosphorylation of key proteins (IL-6 and STAT3) that are involved in signal transduction leading to immune activation and inflammation. The WHO has made a strong recommendation for JAK inhibitors, specifically baricitinib, in patients with severe and critical COVID-19 [219]. Moreover, due to its affinity to a regulator of endocytosis, AP2-associated kinase 1 (AAK1), it has been reported to act as an antiviral by reducing SARS-CoV-2 endocytosis [220].

TLR pathway inhibitors were appearing on the horizon of promising therapeutics after revealing the correlation between high levels of several TLRs and severe COVID-19 [108]. Especially TLR3, TLR4, TLR7, and TLR8 were shown to induce cytokine production, so targeting these specific TLRs could reduce the risk of an hyperinflammatory response during COVID-19 [221]. A Phase II clinical trial of the TLR7/8 inhibitor M5049 (NCT04448756) in COVID-19 patients suffering from pneumonia was completed without providing conclusive data. Similar to other immunomodulatory treatments, TLRs with their key role in innate immunity should not be completely blocked to maintain an antiviral barrier. Nevertheless, targeting specified TLR subtypes with an optimized dose and duration of the treatment could be an effective strategy for the treatment of COVID-19 [222,223]. Additionally, in vitro and in vivo studies have shown that the TLR4 antagonist FP7 significantly reduces the production of lethal lipopolysaccharide (LPS)-mediated cytokines during influenza infection [224].

Although it is debated whether single anti-cytokine therapies are beneficial in COVID-19, the involvement of many cytokines in cytokine storm suggests that the effect of combined therapies may be clinically better [225]. In the current NIH guideline, the recommended treatment for severe COVID-19 patients who need oxygen support is remdesivir plus dexamethasone, and tocilizumab can be added to this treatment in people with more critical disease (hospitalized and requiring ECMO). Depending on the clinical condition and progression of the patient, the treatment regimen can be changed [186,226].

It is reported that ARDS was treated using systemic corticosteroids in France during the 2009 H1N1 influenza pandemic and in China during the 2013 H7N9 avian influenza pandemic. However, there is insufficient evidence to support the use of corticosteroids in severe influenza [227]. The anti-inflammatory and immunosuppressive effects of glucocorticosteroids are based on three mechanisms; (I) the direct effects on gene expression by the binding of glucocorticoid receptors to glucocorticoid-responsive elements, (II) the indirect effects on gene expression through the interactions of glucocorticoid receptors with other transcription factors i.e., NFκB and activator protein 1, and (III) the glucocorticoid receptor-mediated effects on second-messenger cascades [219]. The use of corticosteroids in COVID-19 disease was not routinely recommended by the WHO because it inhibits the immune response, which has a key role in the defense of the host against viruses, reduces viral clearance, and increases the risk of secondary infection. However, multiple randomized studies show that systemic corticosteroid therapy improves clinical outcomes and reduces mortality in hospitalized patients with COVID-19 who need supplemental oxygen by reducing the systemic inflammatory response that is induced by COVID-19 [228,229]. Importantly, this demonstrates again the importance of monitoring the patient’s disease state and clinical parameters. Whereas the described antivirals such as nucleoside analogs should be given to a patient in a rather early phase of the disease for optimal outcome, patients with already progressed disease including a hyperinflammation benefit from immunomodulation such as glucocorticosteroids and also possibly kinase inhibitors and cytokine inhibitors.

## 9. Long-Term Complications of COVID-19

Prolonged disease symptoms of COVID-19, termed as ‘long COVID’, ‘long-haul COVID’, or ‘post-COVID syndrome’, have been reported. The National Institute for Health and Care Excellence (NICE) defined three categories according to persistent symptoms after SARS-CoV-2 infection: (I) acute COVID-19 (for up to four weeks), (II) ongoing symptomatic COVID-19 (four to 12 weeks), and (III) post-COVID-19 syndrome (more than 12 weeks). Long-term complications have also been reported during earlier pandemics, including the influenza pandemics of 1889 and 1892 (Russian flu) and the Spanish flu pandemic in 1918. For instance, post-infectious neurological conditions such as chronic fatigue syndrome that were observed during the Russian flu resemble symptoms described as ‘brain fog’ in patients suffering from long COVID [230]. During the 2009 H1N1 pandemic, the most frequently described symptoms were viral myocarditis and influenza-associated encephalopathy or encephalitis (IAE) [231]. Increased levels of IL-6, IL-8, IL-10, and TNF-α support the hypothesis of systemic inflammation as an important driver for virus-associated myocarditis [232]. Another study of patients with H7N9 infection between March 2013 and March 2014 reported restrictive patterns on pulmonary function as long as two years after discharge from the hospital [233]. Thus, up to now, severe influenza infection can be associated with prolonged complications of lung injury as well as multi-organ dysfunction. Considering that the primary site of SARS-CoV-2 infection appears to be the lung, pulmonary symptoms including prolonged oxygen requirement [234], difficult ventilator weaning [235], fibrotic lung damage [236,237,238], and a reduction in the diffusion lung capacity, are reported after acute COVID-19 infection [236,237,239,240]. Cardiac symptoms such as chest pain were reported in up to 20% of patients that were suffering from COVID-19 at 60 days follow-up [241,242]. Further, cardiovascular magnetic resonance imaging in recovered patients revealed cardiac involvement in 78% and ongoing myocardial inflammation in approximately 60% for more than two months, independent of pre-existing conditions, severity, and overall course of the acute illness [243]. In addition, neurological and neuropsychiatric manifestations including chronic malaise, diffuse myalgia, sleep abnormalities, chronic headache, depression, anxiety, and post-traumatic disorder [234,239,244,245,246] along with cognitive impairments such as difficulties with concentration, memory receptive language or executive function [247,248,249], and venous thromboembolism [250] were reported as a sequelae of acute COVID-19. Further, sequelae including the gastrointestinal system [251], endocrine manifestations such as new-onset diabetes and diabetic ketoacidosis [252], acute kidney injury [44,253], as well as dermatologic sequelae especially hair loss [239,244] are reported. At present, the underlying pathophysiological mechanisms are largely unknown and immunopathological events between acute COVID-19 and post-COVID syndrome have been overlapping. However, there are available hypotheses that could explain prolonged sequelae of COVID-19, elaborating immunologic aberrations and inflammatory damage, induced autoimmunity, and persistence of the virus in certain organs. Especially at infection sites such as the lung or the URT, a delay or defect in the resolution of inflammation may explain the persistence of symptoms. A recent study prospectively analyzed a cohort of 31 individuals with long COVID, comparing them with 31 asymptomatic age and gender matched controls from the same cohort who had prior COVID-19 infection but lacked long COVID symptoms and individuals who had been infected with common cold coronaviruses (HCoV, HCoV-NL63, O229E, OC43, or HKU1). Patients with long COVID demonstrated persistent elevation of IFN-β, PTX3, IFN-γ, IFN-λ2/3 2/3, and IL-6 at months eight, indicating a delayed or defective resolution in the inflammatory response. Further, frequencies of activated CD14^+^CD16^+^ monocytes and plasmacytoid dendritic cells (pDCs) were higher in patients that were suffering from long COVID at eight months after infection [254]. A not yet published study showed a long-lasting cytokine signature in patients with prior COVID-19 where IL-1β, IL-6, and TNF-α showed a significant correlation with post-acute COVID-19. Bronchoalveolar lavage fluid of patients with acute COVID-19 indicated that these cytokines were concentrated in the pro-inflammatory subset of lung macrophages [255]. Further, a follow-up prospective study of 113 patients who developed ARDS during admission for COVID-19 found that 81% had persistent symptoms eight months post-infection suggesting that COVID-19-related ARDS is associated with long-term consequences of COVID-19 [233]. Similarly, at one-year post-ICU discharge, a majority of survivors of (H1N1)-associated ARDS showed lung disabilities as well as psychologic impairment [256]. In addition, impaired immune homeostasis may lead to irreversible pulmonary fibrosis that severely compromises respiratory effector function. Recent single-cell analyses from the lungs of individuals who died of COVID-19 revealed that alveolar type 2 (AT-II) lung epithelial cells failed to undergo full transition into alveolar type 1 (AT-I) cells, indicating an inflammation-associated transient progenitor cell state by AT-II cells [257]. In fact, impairment of AT-II cells could potentially lead to a reduction of tissue-resident macrophage compartment as local maintenance of alveolar macrophages is dependent on the production of CSF-2 by AT-II cells [258]. Another immune dysregulatory response contributing to long-term complications of COVID-19 is the hyper-stimulation of the immune system after SARS-CoV-2 infection that leads to an autoimmune response against self-tissue antigens. A cohort of 194 individuals that were infected with SARS-CoV-2 showed marked increases in Auto-Abs against immunomodulatory proteins [259]. Similarly, protein arrays in serum from 147 hospitalized COVID-19 patients identified Auto-Abs against IFNs, ILs, or other cytokines in approximately 50% of patients, compared with control individuals demonstrating less than 15% [140]. Although the range of long-term complications of COVID-19 is wide, specific high-risk factors are predictive for post-acute COVID-19. Especially the severity of illness during acute COVID-19 has been significantly correlated with long COVID. Patients with more than five symptoms in the first week of acute infection were more likely to develop long COVID symptoms [260]. In addition, patients with moderate to severe acute COVID-19 were more likely to develop ongoing COVID-19 symptoms for 8 to 12 weeks compared to patients with milder disease [261]. Other risk factors for ongoing dyspnea were given by the need for admission to an ICU and the requirement for mechanical ventilation, pre-existing respiratory disease, higher body mass index, older age, as well as Black, Asian, and minority ethnicities [262]. Further, a post-acute COVID-19 study from Wuhan suggested sex differences, with women having a higher probability to encounter fatigue and anxiety or depression six months after symptom onset [239]. Similarly, in a not yet peer-reviewed analysis of 10 longitudinal studies and 1.2 million electronic healthcare records age, female sex, ethnicity, general and mental health, on top of overweight or obesity were associated with a higher risk of long COVID [263]. In addition to the variety of risk factors, the impact of different vaccines on the symptoms of patients that were suffering from long COVID was implemented. An early observational study that was conducted by the LongCOVIDSOS patient advocacy group from the United Kingdom already suggested that long-term symptoms of patients that were suffering from COVID-19 overall improved after vaccination, especially after receiving the Moderna mRNA vaccine [264]. A recent cross-sectional study of patients that were tested between March 2020 and November 2021 from Israel revealed that vaccination with at least two doses of COVID-19 vaccine was associated with a substantial decrease in long-term complications of patients after infection with COVID-19, suggesting that vaccination may have a protective effect against long COVID [265]. One study reported an improvement in CRP levels following a third dose of the COVID-19 vaccine in patients that were suffering from long-term symptoms after breakthrough infection. This suggests that booster immunization might be beneficial to a persistent systemic inflammation. However, the study was conducted only with a small number of samples and without a control group, underlying the need of larger high-quality data in respect to the prevention and treatment of the long-term effects of SARS-CoV-2 infection [266]. In accordance, a large cohort conducted that in comparison to patients with SARS-CoV-2 infection who were not previously vaccinated, patients with breakthrough infection exhibited lower risk of post-acute sequelae [267].

## 10. COVID-19 and Long COVID in Children

In general, SARS-CoV-2 appears to induce milder disease in children compared to adults, which is reflected by low hospitalization numbers of children [268,269]. The cumulative reasons for this finding are not fully understood, but studies suggest that differences in the composition of immune cells and immune gene expression in the URT between children and adults could be involved. While the expression of the viral entry factors ACE2 and TMPRSS2 were similar among different age groups, genes that were associated with IFN-signaling were higher expressed in children [270]. In addition, a significant higher basal expression of the PRRs MDA5, RIG-1, and LGP2 as well as an increased amount of overall immune cells within the URT mucosa could be measured in children compared to the samples of healthy adults, which could provide another layer of protection in the URT of children. In line with these results, a higher increase in PRR gene expression during the early phase of SARS-CoV2 infection among children than adults was also observed in this study [271]. It was also reported that airway epithelial and immune cells in healthy children were already in an IFN-activated state prior to the infection [272]. Altogether, these findings suggest that the mucosal immune system of children facilitates a more vigorous antiviral protection against SARS-CoV-2 infection and disease development. 

Nevertheless, children can develop severe COVID-19 with high morbidity. In May 2020, the South Thames Retrieval Service described an unusual cluster of eight children showing signs of a hyperinflammatory shock syndrome such as the previously described Kawasaki disease within 10 days. A total of four of these patients had previously been exposed to SARS-CoV-2 within their families and two of them later tested positive [273]. The respective disease has now been termed “Multisystem inflammatory syndrome in children (MIS-C) associated with COVID-19” by the Centers for Disease Control and Prevention (CDC) and is sometimes also referred to as pediatric inflammatory multisystem disease syndrome that is temporally associated with SARS-CoV-2 (PIMS-TS) [274]. Different meta-analyses have described the disease symptoms and the patient outcomes of children that were diagnosed with MIS-C. In one study from January 2020 to July 2020, the most common symptoms were fever, diarrhea/abdominal pain, and vomiting with 100%, 73.7%, and 68.3% of all patients affected, respectively. Other frequent symptoms were conjunctivitis and a rash with 51.8% and 56.2%, respectively. Inflammation markers as well as cardiac markers were extremely elevated and 54% of patients had an abnormal electrocardiogram. Whilst 22.2% of patients needed mechanical ventilation and 4.4% extracorporeal membrane oxygenation, only 1.7% of all patients did not survive [275]. A similar meta-analysis from December 2019 to August 2020 also reported that 56.3% of pediatric patients that were suffering from MIS-C presented with shock [276]. The pathogenesis of MIS-C is still not fully understood even though the temporal connection with the onset of the SARS-CoV-2 pandemic makes an association with this new respiratory virus highly likely. Nevertheless, there is some consensus that infectious agents such as bacteria and viruses together with a certain genetic predisposition play a role in the development of Kawasaki disease [277].

Interestingly, during the 2009 H1N1 IAV pandemic, cases of Kawasaki-similar disease were reported in children that were infected with this respiratory virus. According to a recent review, five cases of Kawasaki disease that were associated with H1N1, 21 cases that were associated with non-H1N1 Influenza A viruses, as well as 11 cases that were associated with Influenza B viruses have been reported so far in the literature [278] suggesting that the induced immune responses by these viruses contribute to similar disease outcomes. Even though it is not known whether Influenza virus infection was responsible for the development of Kawasaki disease or only concomitant, these findings raise the possibility that a Kawasaki-similar disease might also accompany future pandemics that are caused by different respiratory viruses. 

Theories for possible disease mechanisms of MIS-C range from changes in the IFN responses to a SARS-CoV-2 infection that results in a cytokine storm to an Auto-Ab-associated pathogenesis that is triggered by an infection with SARS-CoV-2 [279]. Although there is no established treatment protocol for MIS-C, usually intravenous immune globulin treatment (IVIG), glucocorticoids or IVIG plus glucocorticoids are used to treat those children that are affected. An international observational cohort study of clinical and outcome data involving 614 pediatric patients found no significant differences in patient outcome in either of these three most common treatment options [280]. In the future, further research into possible markers that could be predictive for the disease course in children is urgently needed, in order to separate the small number of pediatric patients that are at high risk of developing MIS-C from the majority of children that fortunately seem to have a milder disease course than comparable adult patients. 

To date, little is known about long COVID in children and more research is necessary to establish a clinical case definition and identify objective methods for surveillance. A recent matched, longitudinal cohort study of 11 to 17 years old children and adolescents revealed the rate of physical and mental health long-term complications 3 months after laboratory-confirmed SARS-CoV-2 infection. Interestingly, adolescents who tested positive for SARS-CoV-2 had similar symptoms to those who tested false-negative but were more likely to have multiple symptoms including fatigue, headache, and shortness of breath at the time of PCR testing and three months follow-up [281]. Another prospective cohort study from children between 5 and 17 years estimates the number of symptomatic SARS-CoV-2 infections to be approximately 2.3% with a median duration of six days in older and three days in younger individuals. Importantly, 4.4% of infected children presented ongoing symptoms for 28 days and 1.8% for at least 56 days. The most common symptoms were fatigue, headache, and anosmia [282]. These studies clearly demonstrate that children can also be severely affected by SARS-CoV-2 and experience long-term sequelae. However, the long-term consequences for childhood development as well as potential physical and mental restraints still require more investigation.

## 11. Conclusions

Due to the absence of existing protective adaptive immune responses, infections with the zoonotic viruses SARS-CoV-2 and HPAIV of different subtypes cause severe disease with inflammatory complications in multiple organs. These maladaptive immune responses on top of the virus-induced perturbation of the local antiviral IFN response that would otherwise restrict viral replication and dissemination within the URT and lungs are key factors in the development of COVID-19 but also play important roles during other viral infections such as infections with HPAIV. In addition, the SARS-CoV-2 pandemic has shed light on several unexpected health conditions that represent risk factors for the development of severe COVID-19 and likely resemble the risk factors that are also associated with other respiratory virus infections. In particular, the high prevalence of individuals with a genetically defective type I IFN response as well as pre-existing Auto-Abs against type I IFNs, which render the local antiviral responses inactive and contribute to the harmful dysbalanced cytokine responses, has been an important lesson from this pandemic. The high-prevalence of Auto-Abs against IFN-α and IFN-ω subtypes than the others reinforce the importance of exploring the subtype diversity of type I IFN family to fine-tune antiviral therapy to have maximum efficacy and tolerance. This newly gained knowledge should be recognized and urgently applied for the development of point-of-care testing systems that allow for the detection of such antibodies in a fast and easy manner. Such tests could not only be helpful to improve the prediction of the disease trajectories during COVID-19 or other viral infections directly at the bedside, but could also provide important information for the selection of appropriate and personalized medications. While such strategies could decrease mortality rates during future pandemics, also the risk for long-term complications could be alleviated, which would, next to the positive outcome for the individual patient, result in enormous economic advantages. Last but not least, the development of sequence-adapted recombinant IFNs that circumvent recognition by these Auto-Abs could majorly improve the therapeutic applications of IFN during diverse diseases. Justified by the cytokine-driven nature of COVID-19 and the extremely high number of affected individuals, this pandemic has invigorated clinical investigations on repurposed immunomodulatory therapeutic strategies that target the expression and/or signaling of individual pro-inflammatory cytokines. This has not only resulted in several newly-approved anti-inflammatory therapeutics for COVID-19 treatment by the European Medicines Agency (EMA) and FDA, such as the JNK inhibitor baricitinib (Olumiant, Eli Lilly), the IL-1 receptor antagonist anakinra (Kineret, Sobi), and the IL-6 targeting mAb tocilizumab (Actemra, Roche) but also new direct-acting antivirals, including remdesivir (Veklury, Gilead), nirmatrelvir/ritonavir (Paxlovid, Pfizer), and molnupiravir (Lagevrio, Merck) that can now be administered to patients at early and also late stages of COVID-19 and have greatly improved the chances of survival or onset of severe disease. However, these studies have also revealed the high importance of determining the optimal timing for the application of antiviral or anti-inflammatory treatments, which is especially important for viral diseases such as COVID-19 with an initial phase of viral replication that is susceptible to antiviral treatments, followed by a majorly immune response-driven phase in which the use of antivirals remains without effect and should be replaced with by anti-inflammatory approaches. Naturally, the establishment of reliable biomarkers and disease manifestations to define these phases was and still is a major challenge. It can be assumed that this progress will also be of major advantage for other infectious diseases, especially severe infections with IAV and HPAIV. Beyond repurposing of approved drugs, the pandemic has fostered studies on new approaches, several of which have shown promising results and are still under clinical investigation [283,284].

Although not discussed within this review, one of the most important achievements during this pandemic is the development and approval of several highly efficient SARS-CoV-2 vaccines, in particular the first approved mRNA vaccine, within only one year’s time; however, this has been discussed elsewhere [285]. Similar to the distribution and availability of SARS-CoV-2 vaccines, equal accessibility to these drugs on a global level should be our first priority as part of a sophisticated international pandemic strategy. Next to the improvement of the therapeutic toolbox for the treatment of infectious diseases, the pandemic has increased our attention towards the enormous consequences of long COVID as a chronic disease and its real-life impact. Indeed, one of the most important lessons is that our understanding in science and health research can be enriched by the lived experiences of patients. Substantial patient involvement not only advocates for their illness to be better recognized, researched, and cared for, but also for the prevention of additional people being affected by long COVID. Accordingly, systems that measure recovery and continued illness following SARS-CoV-2 or other viral infection are needed as well as defining and frequently updating clinical case definitions. Without it, people living with long COVID may have considerable barriers in patient healthcare, social care, employment, or financial benefits [286,287].

## Figures and Tables

**Table 1 cells-11-02198-t001:** Summary of the differences and similarities of COVID-19 and HPAIV-induced disease.

	COVID-19	HPAIV (H5N1, H7N9 etc.)
Human-to-human transmission	Yes, very efficient via aerosols Transmission before symptoms onset possible	No (only rare reports) Until today, persistent incompatibility to the human receptor
Cell entry	Primary target cell: Type II pneumocytes, ciliated cells Human receptor: ACE2 Spike processing protease: TMPRSS2, Catepsin L	Primary target cell: Type II pneumocytes Human receptor: α-2,6-Sialic acid HA processing protease: TMPRSS2, Furin
Clinical manifestations	Anosmia, fatigue, cough, dyspnea, sore throat, diarrhea, nausea, pneumonia, ARDS, organ failure	Hospitalized cases: Pneumonia, ARDS, organ failure, leukopenia, lymphopenia, decreased platelets, encephalitis, septic shock
Disease characteristics in severe cases	Target organ: URT, lungs Organ tropism: lung, heart, kidneys, brain, gut Immune response: excessive/dysregulated cytokine levels	Target organ: URT, lungs Organ tropism: lungs, brain, heart, kidneys Immune response: excessive/dysregulated cytokine levels
Biomarkers and laboratory parameters for severe disease	High cytokine levels of IL-6, IL-1β, IL-2, IL-8, IL-17, G-CSF, GMCSF, IP-10, MCP-1, TNF-α Increased levels of CRP, Procalcitonin, LDH, and D-Dimers High ferritin/transferrin ratio Glycocalyx damage	High cytokine levels of IL-6, IL-8, TNF-α, CXCL9, IP-10, MCP-1 Increased levels of CRP, LDH, and D-Dimers High creatinine and aminotransferases
Involved immune receptors	MDA5, TLR1/2/4/5/8/9	RIG-I, TLR3/7, PKR
Monoclonal antibodies approved by FDA/EMA	Sotrovimab, Bebtelovimab, Tocilizumab	
Direct-acting anti-viral drugs (DAA), approved by FDA/EMA (incl. emergency use)	Nucleoside analogs: Remdesivir, Molnupiravir Protease inhibitor: Paxlovid	NA inhibitor: Oseltamivir Endonuclease inhibitor: Baloxavir
Anti-inflammatory treatments, approved by FDA/EMA (for emergency use)	JAK1/2 inhibitor: Baricitinib	
Long-term symptoms	Long COVID: ‘Brain fog‘, restricted pulmonary function, cardiac symptoms, myocardial inflammation, neurological and neuropsychiatric symptoms, venous thromboembolism, gastrointestinal symptoms, new-onset diabetes, diabetic ketoacidosis, acute kidney injury, hair loss	Chronic fatigue syndrome, myocarditis, encephalopathy or encephalitis, restricted pulmonary function, multi-organ dysfunction

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
