# Peer review of "Virus Infection and Systemic Inflammation: Lessons Learnt from COVID-19 and Beyond"

_cells, 2022, doi:10.3390/cells11142198_

Round 1

Reviewer 1 Report

The current manuscript authored by Faist et al. focuses on the responsible mechanisms that SARS-CoV-2 and highly pathogenic avian influenza viruses cause severe disease with inflammatory complications in humans. It is built upon key and up-to-date literature that describes viral infections that lead to either over-activated or delayed local interferon responses, which facilitate efficient viral replication in the primary target organ, systemic viral spread and rapid onset of organ-specific and harmful inflammatory responses. The authors discussed biomarkers for the prediction of disease progression, PRRs and interferon responses during COVID-19 and HPAIV infections, antiviral and immunomodulatory treatments for COVID-19, together with long-term complications of COVID-19.

Overall, the review is nicely written in a balanced way contributing with new aspects of findings about COVID-19 in the past two years. However, I would ask the authors to address several key issues for further consideration.

Major concerns:

1.     Line 93-97. It is well known that common CoV and seasonal IAV mainly infect URT cells. However, for SARs-CoV-2 and avian IAV, the targeted cells are mainly located in LRT, such as AT-II. Hence, they cause more severe diseases and worse outcomes. Please revise this part and make it clear. Also, alveolar macrophages are important in recognition of the two viruses and initiating innate responses to them.  The authors need to discuss the role of alveolar macrophages here and in Line 219-221.

2.     Figure 1 is not cited in the text. There are some problems in the figure. First, TLR3 is one of the PRRs. They should not be drawn side by side. Secondly, there are many abbreviations in the figure that are never mentioned in the text, such as UNC93B1, TICM1. At least, their full names need to be listed in the figure legend. Last, the viruses bind to their receptors (AEC2 or sialic acid) on the cell surface and internalized into the cells. For example, TLR3 recognizes double-stranded RNA in endosomes, which is a common feature of viral genomes internalized by macrophages and dendritic cells. However, the figure shows falsely that the viruses directly bind to the PRRs on the cell surface. The authors should address these issues.

3.     Table 1. Under “cell entry”, studies show that ciliated cells are also the primary target of SARS-CoV-2 in URT. (Ravindra NG, etc. PLoS Biol 2021;19:e3001143) This needs to be added to the table. Meanwhile.under “Biomarkers and laboratory parameters for severe disease”, the authors stated in Line 178. “In addition, high blood levels of IL-1β, IL-2, IL-8, IL-17, G-CSF, GMCSF, IP-10, MCP-1, and TNF-α are indicative for severe COVID-19.” These cytokines should be added in the table besides IL-6.

4.     Too many “as well as” used in the manuscript. Consider replacing them with “along with”, “plus”, etc.  Although the review is mostly written in elegant English, there are a number of grammatical errors, problems in sentence structure and word usage. It is recommended that the manuscript be checked by a native English speaker.

Minor concerns:

Line 35.    Use full name for SARS as it appears the 1st time in the manuscript. Also, please remove “β-CoV” as it brings confusion to readers.

Line 39-41.  Add references to the statement.

Line 54.  SARS-CoV-2 infection and COVID-19.  Same thing.  Use one of them. 

Line 67.  Use full name for ACE2.

Line 71-74. “In order to prevent the collapse of health care systems and interfere with viral transmission, rapid enforcement of non-pharmaceutical interventions such as face masks, strict social distancing, contact tracing and isolation of infected individuals.”  This is not a complete sentence.  Please revise it.

Line 104-105.     “even” might be more appropriate than “already” here.

Line 107.   It would be great if the different symptoms are discussed here for all SARS-CoV-2 variants, Delta, Omicron, etc.

Line 124.  50 % should be 50%. No space.

Line 125.    What dose “a non-fatal case” mean here? Earlier, the authors talked about “H5N1 subtype with a case fatality rate of approx.  50%”.  This new “non-fatal case” could belong to one of non-fatal cases of the 50% of the “H5N1 subtype”?

Line 136.  Remove “both”.

Line 143-157. This paragraph talks about the inflammatory cytokine biomarkers of HPAIV. It is better to move this part after Line 190.

Line 166.   “be a predictive marker” for what? fatal outcome?

Line 213-215.  Add references to the statement.

Line 209.   Add full names for MyD88 and TRIF.

Line 243-245.  Add references to the sentence. “Interestingly, higher expression levels of TLR1, TLR4, TLR5, TLR8, and TLR9 and the TLR-adaptor protein MyD88 were also positively correlated with severity of COVID-19.”

Line 268.  Change “infections, however…” to “infections. However…”.

Line 283-286 &302-304.  Add references to the statements.

Line 418.  It is better to use either SARS-CoV-2 or COVID-19 here.    

Line 453 and 502.  Use subtitle “7.1 Direct-acting antivirals for treatment of COVID-19” and “7.2 Immunomodulatory strategies to alleviate immunopathology during COVID-19”.

Line 456.  Use full name for RNA-dependent RNA polymerase (RdRp) inhibitors.

Line 509.  Remove “,”

Line 601.  Please change “still to date” to up to now or so far.

Line 662.  Please add “to” after likely. The sentence will be “Patients with more than five symptoms in the first week of acute infection were more likely to develop long COVID symptoms.”

Line 682.  Add references to the statement. “….patients after infection with COVID-19, suggesting that vaccination may have a protective effect against long COVID.”

Line 771, Should be “10. Conclusion” not “5. Conclusion”.

Reviewer 2 Report

The authors are to be congratulated for putting together an extremely comprehensive and timely review on lessons learnt from COVID-19 and beyond.

Some minor comments that can help improve the overall manuscript-

1. It would be greatly appreciated if the authors can have a good discussion on  the adaptive immune responses in SARS-Co-V2 infections especially highlighting the lack of timely T cell response.

2. With the increasing knowledge of PRR in COVID-19, describing use of TLR pathway inhibitors as a therapeutic option should be addressed.
